# Changes Associated with Improved Outcomes for Cats Entering RSPCA Queensland Shelters from 2011 to 2016

**DOI:** 10.3390/ani8060095

**Published:** 2018-06-12

**Authors:** Caroline Audrey Kerr, Jacquie Rand, John Murray Morton, Ronelle Reid, Mandy Paterson

**Affiliations:** 1School of Veterinary Science, the University of Queensland, Gatton, QLD 4343, Australia; j.rand@uq.edu.au or jacquie@petwelfare.org.au (J.R.); John.morton@optusnet.com.au (J.M.M.); mpaterson@rspcaqld.org.au (M.P.); 2Australian Pet Welfare Foundation, Kenmore, QLD 4069, Australia; 3Jemora Pty Ltd., Geelong, VIC 3220, Australia; 4Royal Society for the Prevention of Cruelty to Animals (RSPCA), Wacol, QLD 4076, Australia; rreid@rspcaqld.org.au

**Keywords:** cat, shelter, RSPCA, admission source, outcomes, stray, surrendered, adopted, euthanized, desexed

## Abstract

**Simple Summary:**

The aim of this study was to identify changes that contributed to the markedly improved live release of cats in Royal Society for the Prevention of Cruelty to Animals (RSPCA) Queensland shelters by comparing data from 2011 and 2016. Admission numbers were similar in these two years. The number and percentage euthanized decreased substantially from 58% in 2011 to 15% in 2016. The greatest contributor to this were marked increases in cats rehomed (i.e., adopted). The number of cats adopted doubled from 2011 to 2016, with over half the increase contributed by increased shelter adoptions, and the remainder achieved by increased off-site adoptions, largely through agreements with Petbarn stores. Improved outcomes were facilitated by nearly doubling the number of cats temporarily in foster care. Cats euthanized for behavioral reasons decreased by 85%, including a marked decrease in the number of euthanasias because the cat was deemed feral. Euthanasia of young kittens dramatically decreased. The number of cats reclaimed by their owner was similar in the two years and was only a small contributor to the numbers of cats released live. To achieve further improvements, programs that decrease intake for both stray and owned cats would be beneficial.

**Abstract:**

This retrospective study of cat admissions to RSPCA Queensland shelters describes changes associated with improved outcomes ending in live release in 2016 compared to 2011. There were 13,911 cat admissions in 2011 and 13,220 in 2016, with approximately 50% in both years admitted as strays from the general public or council contracts. In contrast, owner surrenders halved from 30% to 15% of admissions. Percentages of admissions ending in euthanasia decreased from 58% to 15%. Only 5% of cat admissions were reclaimed in each of these years, but the percentage rehomed increased from 34% to 74%, of which 61% of the increase was contributed by in-shelter adoptions and 39% from non-shelter sites, predominately retail partnerships. The percentage temporarily fostered until rehoming doubled. In 2011, euthanasias were most common for medical (32% of all euthanasias), behavioral (36%) and age/shelter number (30%) reasons, whereas in 2016, 69% of euthanasias were for medical reasons. The number of young kittens euthanized decreased from 1116 in 2011 to 22 in 2016. The number of cats classified as feral and euthanized decreased from 1178 to 132, in association with increased time for assessment of behavior and increased use of behavior modification programs and foster care. We attribute the improved cat outcomes to strategies that increased adoptions and reduced euthanasia of young kittens and poorly socialized cats, including foster programs. To achieve further decreases in euthanasia, strategies to decrease intake would be highly beneficial, such as those targeted to reduce stray cat admissions.

## 1. Introduction

Large numbers of stray and owned cats are admitted to animal shelters annually, and historically have had poor survival outcomes [1,2,3], with euthanasia percentages reported to be in excess of 58% in 2010 in Australia [4] and 50% to 70% in the USA [2,5,6]. As the animal welfare and social impact of this is substantial [7], animal welfare agencies in Australia and internationally have made considerable efforts to improve outcomes [2]. As a consequence, euthanasia percentages reported by the major welfare agencies in USA and Australia decreased to 41% in 2014/2015 [8] and 29% in 2015/2016 [3], respectively. However, achieving further decreases in euthanasia of cats entering shelters is an ongoing and complex challenge. Consequently, research into understanding what strategies are effective in decreasing euthanasia of cats would be advantageous in defining strategies to reduce the number of cats euthanized in shelters. 

One way to reduce the number of cats euthanized in shelters is by reducing intake through addressing the factors that lead to cat admissions. Desexing programs [9] have traditionally been utilized to reduce numbers of stray and owned kittens and cats entering shelters [10,11,12], and their euthanasia [5,13]. Other strategies used by shelters may include assisting owners to keep their cat rather than surrender it, by providing advice and support, especially for medical and behavioral issues (called ‘pet diversion’ or ‘pet retention’ programs) [14]. The number of cats euthanized can also be reduced by increasing the number of cats that are reclaimed by owners and adoptions. Programs that increase the number of owned cats with microchips, and increase the percentage of owners with accurate contact details , assist in increasing the number of lost cats that are reclaimed in Australia, USA, and the UK [15]. Nevertheless, reclaim percentages are very low at 2% of admissions in USA [16] and 5% in Australia [3]. 

In Australia in 2016, the estimated number of owned cats was just under 3.9 million [17]. The percentage of households with cats increased by 6 percentage points from 23% in 2014 to 29% in 2016 [17], with 25% of owned cats acquired from shelters and rescue groups [17]. In parallel, Royal Society for the Prevention of Cruelty to Animals (RSPCA) Australia’s shelters increased the percentage of cats rehomed from 29% (19,004 cats) in 2010/2011 to 56% (31,178 cats) in 2015/2016. In the USA, 57% of admissions (711,270) were rehomed in 2016 across 2225 shelters [8] and maximizing the number adopted is important in decreasing numbers euthanized [18]. Because increased time in shelters leads to chronic stress and illness, and potentially a greater risk of euthanasia [19], it is crucial that adoption factors and pathways to adoptions are further examined, so that times to adoption can be minimized. Consequently, a number of studies in U.S.A. have examined factors that increase the probability of admitted cats being adopted [19,20,21,22], such as ‘friendliness’ [19] and the degree of play activity displayed by a cat [22]. 

The RSPCA is Australia’s largest animal welfare agency that receives stray and owner-surrendered cats [23]. In 2015/2016, reported euthanasia percentages for the RSPCA varied substantially between states, from 44% in New South Wales (NSW) to 12% in Queensland in 2015–2016 [23], and Queensland reported the greatest decrease over the previous 5 years [3]. Therefore, evaluating intake and outcome data to understand factors associated with the reduction in numbers and percentages of admissions in which the cat was euthanized, and the consequent increase in live release numbers and percentages in Queensland, may assist in improving cat outcomes in other shelters. The aim of this study was to identify changes associated with improved outcomes for cats at RSPCA Queensland in 2016 compared to 2011.

## 2. Experimental Section 

### 2.1. Study Design and Data Collection

A retrospective study was conducted using RSPCA Queensland data for cat admissions to their shelters for the calendar years of 2011 and 2016. Cats were either admitted directly into the RSPCA Queensland shelters (9 shelters in 2011 and 10 shelters in 2016) or other RSPCA Queensland locations (two RSPCA pet shops, one adoption center, and one Friends group both years), or via council contracts (five council contracts in 2011 and six in 2016) (Table A1). Councils in Australia are local government bodies (similar to counties in USA) and are responsible for domestic animal control. Over the study period, stray animals under council control that were not reclaimed within a minimum holding period (typically 72 h in Queensland) could be rehomed. In Australia, holding periods are specified by local councils, and may vary between councils. If there was a contract in place between the council and RSPCA Queensland, cats were moved to the relevant RSPCA shelter, which was adjacent to the council pound (the council animal holding facility). Within other council areas with no RSPCA contract, selected cats were regularly transferred from the council pound to the RSPCA for rehoming.

For cats admitted to RSPCA Queensland facilities, rehoming of unclaimed stray cats only commenced after the mandated minimum holding period of 72 h, and after a cooling-off period of 24 h for owner-surrendered cats. Non-reclaimed cats were assessed behaviorally and medically, and if deemed suitable for adoption, were desexed, and attempts at rehoming commenced either directly from RSPCA Queensland shelters (either the shelter to which the cat was admitted or another RSPCA shelter where the cat had subsequently been moved to) or after movement or transfer to adoption centers, pet shops, or off-site adoption events. The term ‘movement’ is applied to cats staying under the care of the RSPCA, while the term ‘transfer’ refers to cats whose care became the responsibility of the receiving organization. Adoption centers were operated by either RSPCA Queensland or by another organization (Northside Vet Care Outreach Adoption Centre in 2011), and they only rehomed animals, i.e., the full services of a shelter were not available. ‘Pet shops’ were both pet supply stores that RSPCA Queensland owned (e.g., World for Pets) and stores with agreements with RSPCA to stock only RSPCA Queensland animals (one Petbarn and one Pet Crazy store in 2011, and 39 Petbarn stores in 2016). Adoption events were one-day events (‘Pop-Up Adoption Event’ and ‘Big Adopt Out’) operated by RSPCA Queensland and held at external function centers. No adoption events were held in 2011, and 2 were held in 2016. Cats from RSPCA that were not adopted on the day were moved back to RSPCA Queensland shelters for rehoming. The Pop-Up Adoption Event only included cats admitted to and under the care of RSPCA Queensland, while the Big Adopt Out was an event in which rescue groups could also bring cats for adoption, and those cats remained under the ownership and care of the rescue group. Cats contributed by rescue groups to the Big Adopt Out were not included in RSPCA Queensland statistics, nor in the current study data. 

In 2011, RSPCA Queensland’s headquarters were located at the Fairfield shelter. This shelter was closed in December 2011, and the headquarters moved to the Wacol Animal Care Campus, which started to admit cats in late 2011. From December 2010 to the end of January 2011, there were major flooding events that affected a large proportion of the areas of the state in which RSPCA Queensland operated. RSPCA Bundaberg functioned as a flood emergency center and a flood evacuation center temporarily operated at a university site at Rockhampton (under the management of RSPCA Queensland).

RSPCA Queensland admission records were exported from ShelterMate©, RSPCA Queensland’s data management system [24] on 23 March 2017, and imported into Microsoft Excel for manipulation. Data obtained for each admission included cat identification number, name of shelter the cat was admitted to, date of admission, date of birth, and admission source. Admission sources were ‘stray’, ‘owner surrender’, ‘council contracts’, ‘transfers in’ (from non-contracted councils, veterinary surgeries and rescue groups), ‘ambulance’ (RSPCA’s animal ambulance service), ‘humane officer’ (cats seized by inspectors appointed under Queensland animal welfare legislation, with powers to investigate cases of animal cruelty and to enforce animal welfare law), ‘euthanasia request’ and ‘euthanasia request with consent to rehome’ (cats surrendered to RSPCA specifically for euthanasia, but for the latter category—only available in 2016—the owner gave permission for the cat to be rehomed if this was possible and appropriate), ‘return’ (cats adopted but returned within 30 days to RSPCA as unsuccessful adoptions), ‘shelter offspring’ (cats born in care), ‘dead on arrival’, ‘emergency boarding’, ‘bequest in shelter’ (cats left in our care by someone’s estate), and ‘evacuation’ (cats looked after by RSPCA Queensland during an emergency). Before their admission was resolved, some cats were temporarily placed with carers in their homes, referred to as being in ‘foster care’.

Each admission was classified by outcome as reclaim (i.e., cat returned to owner), rehome (adoption to new owner), transfer (to rescue group that rehomes animals), euthanasia (humanely killed), died whilst in care, or ‘other’ (e.g., escaped or stolen). The reason for euthanasia was recorded at the time of euthanasia (e.g., cat flu), and only one reason was recorded. Admissions where the cat was still the responsibility of the RSPCA (including cats in foster care) were classified as being ‘in-care’. Rehoming was further classified as occurring from RSPCA Queensland shelters, RSPCA Queensland-operated pet stores, commercial pet stores (through partnerships between RSPCA and organizations such as Petbarn), and RSPCA-organized adoption events, and through adoption centers (either RSPCA Queensland or independently operated). Some cats were moved between sites so their admission site differed from their outcome site. These movements were to either adoption-only centers, including pet supply stores, or to the RSPCA headquarters shelter because of its greater adoption capacity (especially for kittens) and better medical and behavioral resources. For cats moved between RSPCA sites after admission, the outcome occurred at that subsequent site, but it is shown as an outcome for the admission site in relevant tables. Admission and outcome statistics reported in our study differ slightly from those reported publically [23] due to differences in inclusion criteria, including cats surrendered for euthanasia, and use of calendar year rather than financial year for analyses.

### 2.2. Statistical Analyses

The unit of analysis was the individual admission, where a cat entering an RSPCA Queensland shelter constituted one admission. Each admission ended when the cat was reclaimed, rehomed, `transferred to a rescue group, euthanized, died before release, or had another outcome (e.g., escaped or stolen). Any particular cat could have multiple admissions. Three populations were used for analyses: (1) All admissions were used for descriptive analyses; (2) When describing outcomes by the 90th day of the admission (‘day 90’, in which day 1 was the day the cat was admitted) for all admissions, cats whose outcome was not known by that day were not used; (3) A third, more restricted, population was used for more detailed analyses of percentages of admissions in which the cat was rehomed, using admissions in which the cat was available to be rehomed. For this latter population, the following admissions were excluded: ambulance euthanasia requests, bequests (Pet Legacy Program [25]) and emergency boarding, and admissions in which the cat was reclaimed, as these cats were not available for rehoming. From this latter population, we also excluded admissions not resolved by day 90, and repeat admissions of the same cat within the same calendar year (to minimize any clustering of outcome by admission within cat). Owner surrenders requesting euthanasia were not offered the option of consent to rehome in 2011, but this option was offered in 2016. All euthanasia requests, with and without consent to rehome, were included for both years for consistency. Using cats that were available to be rehomed, probabilities of an admission ending in the cat being rehomed by day 90 were compared separately by age at admission and admission source, using generalized linear models with the log link and binomial error distribution. Iterated reweighted least-squares (IRLS) optimization of the deviance was used. Models were fitted using the -binreg- command in Stata (version 15, StataCorp, College Station, TX, USA). Exponentiated coefficients from these models were interpreted as risk ratios. Year and either age category at intake or animal admission source were fitted, along with interaction terms with year. The overall *p*-values for the interaction terms were assessed using joint Wald tests, using Stata’s ‘-testparm-’ command. For calculation of numbers of admissions per 1000 human residents, human population sizes in 2011 and 2016 for the local government area (council) in which the shelter operated were used [26].

## 3. Results

### 3.1. Number and Source of Admissions

The number of cat admissions (‘admissions’) to RSPCA Queensland shelters decreased by only 5% from 2011 to 2016, from 13,911 to 13,220 (Table 1). There were also minimal changes in the numbers and percentages of admissions contributed by two of the three largest sources of cats: ‘stray’ (4295 versus 4144, 30.9% versus 31.3% of admissions in 2011 and 2016, respectively), followed by ‘council contracts’ (2715 versus 2492, 19.5% versus 18.9%) (Table 1). In contrast, the number and percentage of admissions that were from ‘owner surrender’ were halved from 2011 to 2016 (from 4221 versus 2075; 30.3% to 15.7%). The number and percentage of admissions in which the cat was ‘transferred in’ from non-contract councils, veterinarians, and rescue groups increased from 679 (4.9% of admissions) in 2011 to 1553 (11.7%) in 2016. The number and percentage of ambulance admissions almost doubled from 2011 to 2016 (from 682 and 4.9% to 1209 and 9.2%, respectively) (Table 1).

#### 3.1.1. Incoming Regions of Admissions

For both years, the organizational headquarter shelter was the single largest source of admissions (Table A1). In 2011, this was Fairfield (3653, 26.3% of admissions in 2011) and in 2016, Wacol Animal Care Campus (3333, 25.2% of admissions in 2016). Approximate numbers of admissions per 1000 human residents varied between shelters across the state and over time and ranged from 1 (Noosa) to 36 (Gympie) in 2016. During extensive flooding in 2011, RSPCA Bundaberg functioned as a flood emergency center, and the number of admissions was nearly four times higher in 2011 than in 2016 (1631 versus 464).

#### 3.1.2. Admissions by Time of Year

In 2011, monthly numbers of cat admissions (expressed as percentages of the year’s admissions) peaked in March (1557, 11% of the year’s admissions), following a substantial drop in February, likely associated with widespread flooding in Queensland. Numbers of admissions per month were less variable in 2016 with the highest percentages of the year’s admissions occurring in January, April, and November, each with approximately 10% of admissions (Figure 1). However, for both years, numbers of admissions were lowest in late winter and early spring (i.e., August and September in 2011 and July to September in 2016, with 6–7% of the year’s admissions in each of these months) and were highest in the warmer months (November to May, with 8–11% of the year’s admissions in each of these months).

#### 3.1.3. Admissions by Desex Status, Feral Status, and Age

Desex status was recorded for 86% (11,972) and 95% (12,533) of admissions in 2011 and 2016, respectively (Table 2). Of cats whose desex status was known, most were not desexed on admission—77% (9165) in 2011, and 73% (9116) in 2016. The cat’s date of birth was not recorded for 5607 admissions (40%) in 2011 and for 1496 admissions (11%) in 2016. Of the 6217 admissions in 2016 known to be adult cats (≥17 weeks of age) of known desex status, 42% (2642/6217) were desexed, and the proportion was higher for owner-surrendered (58%; 668/1,157) compared with stray (29% or 472/1628) and council (28%; 357/1296) admissions. There was a small change in the proportions of admissions that were female from 2011 (53%) to 2016 (50%).

RSPCA Queensland classification of feral status was mainly based on the cat’s behavior and absence of identification; cats with no identification, exhibiting non-social behavior, and not settling in their cage within 24–72 h were usually classified as feral. The number and proportion of admissions classed as feral on entry to the shelters decreased substantially from 8% (1154) of admissions in 2011 to 1% (172) in 2016 (Table 2).

Of admissions in which the cat’s date of birth was recorded (8304 and 11,724 in 2011 and 2016, respectively), 57% (4752) were kittens (i.e., aged <17 weeks) in 2011, declining to 46% (5403) in 2016 (Table 3). The largest categories by age in 2011 were ‘4 to <7 weeks’ and ‘7 to <17 weeks’ (each 23% of all admissions); these decreased in 2016 to, respectively, 15% and 21% of admissions in that year. In contrast, the percentages of admissions for every other age category were greater in 2016. Cats ‘7 years or older’ comprised the lowest percentages and numbers of admissions for both years (4% or 311 and 5% or 627, in 2011 and 2016, respectively). In 2016, for cats of known age, the main sources of kittens <17 weeks were stray (37%; 2010 of 5403 kittens), council (17%; 911), transfer in (16%; 841), owner surrender (16%; 888), and ambulance (6%; 323).

### 3.2. Outcomes

#### 3.2.1. General

Admissions were resolved by day 90 for 13,252 and 12,228 admissions (95% and 92% of all admissions) in 2011 and 2016, respectively (Table A2). Outcomes for these admissions are reported in Table 4, Table 5 and Table 6 and Figure 2. Results by site are reported in Table A3, Table A4 and Table A5.

#### 3.2.2. Reclaim

Reclaim percentages were very low and similar across both years. Only 5% of all admissions resolved by day 90 were reclaimed in each of 2011 and 2016 (644/13,252 and 663/12,228, respectively; Table 4). Excluding admissions that were owner surrenders, percentages of admissions that were reclaimed were 6.4% (586/9177) in 2011 and 6.3% (644/10,299) in 2016.

#### 3.2.3. Rehomed

Of admissions in which the outcome was known by day 90 of admission, the percentage ending in the cat being rehomed more than doubled from 34% (4510) in 2011 to 74% (8996) in 2016 (Table 4).

The number of RSPCA Queensland sites rehoming cats increased by 2 when comparing 2011 to 2016. Also, the headquarters shelter site moved from Fairfield to Wacol at the end of 2011. In both years, the organization headquarters sites rehomed the largest number of cats: Fairfield in 2011 (1394, 39.6% of admissions resolved by day 90 were rehomed) and Wacol in 2016 (1826, 58.1%, Table A3 and Table A4). There was a marked increase in the number of commercial pet supply shops operating as cat adoption centers from two stores in 2011 to 39 stores in 2016. Accordingly, substantially more cats were rehomed through commercial stores, with the number rehomed from Petbarn increasing from 86 to 1503 cats; the latter represented 17% of all cats rehomed in 2016 (Table A5). One-day adoption events (‘Pop-up Adoptions’ and ‘Big Adopt Out’) occurred only in 2016, and 387 cats were rehomed from those two events (Table A5). Importantly, the number of admissions in which the cat was rehomed through RSPCA sites increased from 4272 to 6999, a 64% increase, and the number rehomed from non-shelter sites increased from 238 to 1997, a 740% increase, and this latter method of rehoming accounted for 22% of all admissions rehomed in 2016 compared to 5% in 2011 (Table A5). Thus, of the increase in number of admissions rehomed of 4486 (i.e., from 4510 to 8996; Table 4), 61% (2727/4486) was due to an increase in numbers of admissions rehomed from shelter sites and 39% (1759/4486) was due to increased numbers of cats rehomed from non-shelter sites (Table A5).

Of the 13,911 admissions in 2011 and 13,220 in 2016, 89% (12,372) and 83% (10,954), respectively, were used for more detailed analyses of rehoming by day 90; these were admissions in which the cat was available to be rehomed excluding repeat admissions of the same cat within the same calendar year. Percentages of these admissions in which the cat was rehomed by day 90 increased from 2011 to 2016 for all admission sources (Table 1). The greatest increase was for ‘council’ from 29% of council admissions (707) in 2011 to 82% (1766) in 2016 (Table 1). Effects of source differed significantly by year (*p* for interaction <0.001), but risk ratios were similar within each year, so, for simplicity, the pooled effect of source across the two years was reported. Relative to ‘owner surrender’, admissions from all sources other than transfer in had a lower chance of being rehomed, with ‘euthanasia request’ and ‘ambulance’ the lowest (risk ratios (RRs) 0.19 and 0.47, respectively; Table 1). 

Of the admissions used for analyses of rehoming by day 90, 395 were euthanasia requests in 2011 and 199 in 2016. In 2016, the owner gave consent for rehoming if possible and appropriate for 91 of these 199 admissions, while for the remaining 108 admissions, no such consent was given. All of the 395 admissions in 2011 and the 108 admissions in 2016 were euthanized, while, of the 91 admissions in which the owner gave consent for rehoming, 61 (67%) were rehomed, 1 was transferred to a rescue group, 3 died before release, and 26 were euthanized. Collectively, these results show that the improved rehoming percentage in 2016 was not due to more admissions from sources that are more likely to be rehomed. On the contrary, the improved rehoming percentage in 2016 was achieved despite a lower proportion of admissions being from owner surrender or transfer in, the two sources with the highest rehoming percentages. In 2016, 28% of admissions were from one of these sources compared to 35% in 2011 (Table 1).

Of admissions in which the cat was available to be rehomed, all age group categories had higher percentages of admissions rehomed by day 90 in 2016 compared to 2011 (Table 3). The largest increase was in cats aged 7 years or older, from 17% to 44% (Table 3). Effects of age differed significantly by year (*p* for interaction <0.001), but risk ratios were similar within each year so; for simplicity, the pooled effect of age across the two years was reported. Cats aged 4 weeks to <7 years had highest rehoming percentages. The percentages of admissions in which the cat was in this age range were similar in 2011 and 2016 (86% and 85%, respectively). Thus, these results show that the improved rehoming percentage in 2016 was not due to more admissions from age categories that are more likely to be rehomed.

Of all admissions in which the cat was available to be rehomed, there were similar increases in the percentages of admissions in which the cat was rehomed by day 90 for female and male cats in 2011 and 2016 (Table 2), and within each year, the percentage rehomed was very similar for female and male cats. For example, in 2016, 82% of female and 77% of male cat admissions were rehomed. Whether cats were entire or desexed on admission also made little difference to the percentage that were rehomed. For example, in 2016, 78% of desexed and 82% of entire cats on admission were rehomed by day 90 (Table 2). Percentages of admissions where the cat was rehomed by day 90 varied by month of admission from 26 to 45% of admissions in 2011, and from 69 to 81% in 2016, with admissions in December having the highest percentage rehomed in both years (Figure 1). Percentages rehomed generally increased continually through 2011.

#### 3.2.4. Foster

The number of admissions where the cat was fostered almost doubled from 2747 out of 13,911 admissions in 2011 to 4732 out of 13,220 admissions in 2016 and increased from 20% to 36% of admissions. Approximately half of the cats that were fostered were moved to foster care by day 3 for both years, and for 2579 (94%) and 4461 (94%), the cat was fostered by the 30th day after admission. Of those, where the admission was resolved by day 90, percentages where the cat was rehomed by day 90 were similar for both years (87% and 88% for 2011 and 2016, respectively; Table 5).

#### 3.2.5. Transferred to Rescue Groups

A very small number of cat admissions were transferred to rescue groups by day 90 in both years, and this increased marginally from 206 to 279 admissions (2.3% of all admissions that were resolved by day 90) in 2016 (Table 4). Of transfers to rescue groups by day 90 from RSPCA Queensland, the headquarter sites had the highest numbers: 65 (1.8% of admissions resolved by day 90 were transferred to rescue groups) for Fairfield in 2011, and 120 (3.8%) for Wacol Animal Care Centre in 2016.

#### 3.2.6. Euthanized

Numbers of admissions that ended in euthanasia by day 90 substantially decreased from 7656 in 2011 to 1826 cats in 2016, representing a decrease from 58% to 15% of admissions that were resolved by day 90 (Table 4). Euthanasias predominantly occurred by day 8 after admission in 2011 (84% of all admissions were euthanized by day 90), whereas more euthanasias were delayed in 2016 (day 25 to reach 84% of all admissions euthanized by day 90, Figure 2). In 2011, the three most common reasons for euthanasia were ‘behavioral and feral’ (36% of euthanasias), ‘medical’ (32%), and ‘age/space limitations’ (30%) (Table 6). However, in 2016, there was a marked decrease in the percentage of euthanasias that were for ‘behavioral and feral’ (22%), and ‘age/space limitations’ reasons (2%), with most euthanized being for ‘medical’ reasons (69%). Numbers of cats euthanized for behavioral reasons (including feral cats) decreased from 2771 to 393. About 44% of this reduction of 2378 cats was due to fewer euthanasias because the cat was deemed feral, with the number decreasing from 1178 to 132 (Table 6). Of the 13,252 and 12,228 admissions resolved by day 90 in 2011 and 2016, respectively, for 8.9% and 1.1%, the cat was euthanized because it was deemed to be feral. Numbers of cats euthanized for ‘age/space limitations’ reasons also significantly decreased, from 2303 cats to 32 cats in 2016; over half of this reduction was due to fewer kittens aged less than 4 weeks being euthanized (1309 to 15). The number of cats euthanized for medical reasons decreased nearly by half (2454 to 1252), but constituted a bigger percentage (69%) of those euthanized in 2016. The number of cats euthanized based on humane grounds more than doubled from 434 cats to 971 cats, which constituted the largest single category of cats euthanized for medical reasons in 2016. Similarly, the number of feline immunodeficiency virus (FIV) positive cats that were euthanized increased markedly from 87 to 316 cats and constituted 17% of all cats euthanized in 2016.

In 2016, percentages of cats euthanized by day 90 varied by site from 3.4% to 25% (Table A3). Although the headquarters site (Wacol) had the highest euthanasia percentage in 2016 (2.1 times that for other RSPCA sites pooled), after adjustments simultaneously for both age category and source of admission, using a generalized linear regression model as described above, but also with standard errors adjusted for clustering of admission within cat, for admissions to Wacol, the cat was only 1.1 times more likely to be euthanized than admissions to other RSPCA sites pooled. Therefore, the higher percentage euthanized at Wacol relative to other RSPCA sites pooled was almost entirely accounted for by source (particularly the much higher proportion of admissions that were ambulance admissions and the much lower proportion that were transfers in from other organizations) and differences in age category (particularly the higher proportion of admissions that were old (7 years or older) cats and the much lower proportion of admissions that were aged 7 w to 1 year), with source accounting for most of the difference.

#### 3.2.7. Live Release

There was a substantial increase in the number and proportion of cat admissions that were released alive by day 90, that is, reclaimed, rehomed, or transferred to rescue groups by day 90, from 5360 (40% of the 13,252 admissions that were resolved by day 90) to 9938 (81% of 12,228 admissions; Table 4). The proportion in which the outcome was resolved by day 90 varied between RSPCA shelters from 83% to 98% (Table A2).

#### 3.2.8. Movements between RSPCA Sites

Cats were moved between RSPCA sites (i.e., their outcome site was an RSPCA site but differed from their admission site) more commonly in 2016. In 2011, of the 13,255 admissions with both admission and outcome sites recorded and resolved by day 90, for 778 (5.9%) the cat was moved internally compared to 1351 (11.1% of 12,228 admissions) in 2016. These movements were to a range of sites in 2011. In 2016, most (82.2% or 1110) were to the headquarters shelter for adoption; movements commonly involved kittens, and cats moved for medical or behavioral treatment at the headquarters shelter.

## 4. Discussion

The aim of this retrospective study was to identify changes that contributed to the markedly improved outcomes for cats in RSPCA Queensland shelters by comparing 2016 to 2011. The greatest contributor to the increase in number of admissions ending in live release was a marked increase in the number and proportion of admissions in which the cat was rehomed (i.e., adopted). The numbers of cats adopted in 2016 was double that in 2011, with over half of the increase contributed to by increased shelter adoptions, and the remainder of the increase achieved by increased numbers of off-site adoptions, largely through agreements with Petbarn stores. These improved outcomes were associated with nearly a doubling in the number of admissions in which the cat was fostered. The number of cats euthanized for behavioral reasons decreased to one seventh, and number of kittens euthanized decreased from 1305 to only 22 in 2016. The total number of admissions was only marginally less in 2016, and the number of cats reclaimed by owners was similar for the two years and was only a small contributor to the number of admissions ending in live release. 

### 4.1. Cat Admissions

There was small decrease in cat admissions from 2011 to 2016 (13,911 to 13,220 cats), although numbers in both years were lower than the average annual intake of 16,868 cats reported for 2006 to 2008 for RSPCA Queensland [4], and lower than the mean intake of 15,861 cats from 1999 to 2016 (SEM ± 681) [3]. There was marked variation between sites in approximate intake per 1000 residents (from 1 to 36 in 2016). For all sites with admissions greater than 10 cats/1000 residents, the RSPCA had the contract for the council pound and was the main organization in the area accepting cats. The average annual intake for shelters and pounds combined in Australia was reported to be 7 admissions/1000 residents based on 2011–2012 data [27]. Not surprisingly, in the USA, annual numbers of admissions are closely correlated with number of cats euthanized in shelters and municipal facilities [28]. To further reduce euthanasia, locations where RSPCA admission numbers are high warrant substantial resources to decrease the numbers of stray and owner-surrender admissions.

#### 4.1.1. Sources of Admission

In both years, strays found by the general public were the largest source of cat admissions into RSPCA Queensland shelters (approximately 31%), followed by admissions via council contracts (approx. 19%); these are also all strays (but not received directly by the RSPCA from the public). Many of the transfers-in (which constituted 12% of admissions in 2016) and nearly all ambulance admissions (9% of admissions in 2016) were presumably also stray, suggesting at least 50%, and potentially up to 70%, of cat admissions in 2016 were strays. These results are comparable to a 2006 to 2008 RSPCA Queensland study that reported 54% of cats admitted were strays from either the public or council contracts [4]. Similarly, in a Melbourne-based shelter study in 2009, the majority of cat admissions (82%) were strays [29]. 

Unlike dogs, cats admitted to RSPCA shelters do not undergo a formal behavioral assessment. However, if they display behavioral traits that may affect their suitability for rehoming, they are referred to the RSPCA Queensland Animal Rehabilitation Team for assessment. Most of the admitted stray cats were socialized to people, as evidenced by the low euthanasia rate for behavioral reasons. The urban stray cat population consists of abandoned, wandering, or lost owned cats, unowned cats and cats that can be classified as semi-owned [30,31]. Semi-owned cats are provided with care, typically food, by people who care for the cat but do not perceive themselves to be the owner. Previous Australian studies have found that 22% of households fed a cat that was not their own [32], and 9% of respondents to an internet survey fed unowned cats on a daily basis [33]. The majority of these semi-owned cats were not desexed, and compared with owned cats, a greater proportion had produced kittens in the study [32]. In USA, it is estimated in urban areas there are at least 60 semi-owned cats per 1000 residents [30,31,34,35], and semi-owned cats are thought to constitute a large proportion of shelter admissions of stray cats in USA and Australia [29,33,36,37,38].

Substantial reductions in numbers and percentage of cats euthanized are being achieved in USA, where desexing programs for stray and semi-owned cats are used to decrease admission numbers of stray cats into shelters [38]. In one study, when a high proportion of stray cats were desexed in the target area (>50%), numbers of cat admissions to shelters in the area decreased from 13 to 4 cats/1000 residents over two years compared to a decrease from 16 to 14 cats/1000 residents in the non-target area, and numbers of cats euthanized decreased markedly from 8 to 0.4 cats/1000 residents [37]. In community cat desexing programs (in the USA also called diversion programs), stray cats are either trapped or identified as unsocialized within a shelter and desexed, and clinically healthy cats are returned to their home location (trap, neuter, and return or ‘TNR’; and shelter, neuter, and return or SNR). The return of desexed, unsocialized cats to their home range is based on the premise that they may be semi-owned, or even wandering owned cats [39]. Evidence suggests that 75% of lost cats are found within 500 meters of their home [40], and cats are 7 to 13 times more likely to return home than be reunited with their owner from the shelter [41]. In most TNR programs, kittens and friendly adults are usually removed for adoption. For shelters to achieve euthanasia of ≤5–10% for cats, these programs will likely be important, given the substantial proportion of kitten and adult cat intake that is strays. However, releasing unowned cats into the community (whether desexed or not) is currently illegal without a permit in most Australian states, because of legislation relating to abandonment of cats or feral pest legislation [36]. Australian research is required to determine the efficacy of these programs to decrease numbers of stray cats in urban areas, to assess their impact on shelter intake and numbers euthanized, and to evaluate the impact on local wildlife ecology. 

Ambulance admissions nearly doubled due to more ambulances on the road and the use of volunteer drivers (meaning the ambulances were operating more hours). The area serviced by RSPCA ambulances also increased to include the Ipswich council area. After implementation of a community cat desexing program in Albuquerque, USA, calls to municipal services for dead cats in the city declined by 45% [42]. Therefore, numbers of injured cats requiring ambulance services would also be expected to decrease.

The annual number of owner-surrendered admissions halved from 4221 to 2075 cats from 2011 to 2016. This may have been due to surrender diversion strategies implemented by RSPCA Queensland in 2016, in which owners intending to surrender their cat were assisted in retaining their pet by providing help and advice towards resolving the underlying reasons prompting surrender, or were assisted in rehoming it themselves without admission to RSPCA. These strategies included an RSPCA Queensland web page to help cat owners access resources to resolve underlying issues leading to surrender, including medical and behavioral problems [43]. The website also provided information on how owners can rehome their pet and had links to reputable rescue groups. Commencing in 2012, owners were also required to have an appointment with RSPCA staff prior to relinquishing their pet to discuss options to deter impulse surrenders. In the USA, programs aimed at reducing numbers of shelter admissions by supporting owners with counselling for behavior problems and financial assistance for health problems and cat licensing costs (not required in most parts of Queensland) are effective at decreasing the number of cats surrendered by owners [41]. In one US report, 70% of owners offered support elected to keep their pet, and the estimated average cost of averting a pet from being admitted to a shelter was $US50 [44,45,46]. Therefore, such programs are likely to be cost-effective, given that shelter costs in Australia for a cat rehomed after one week were estimated at $756 (cat admission, basic health care, desexing, housing for one week, etc.), plus, for cats for whom rehoming takes longer than one week, costs include a further $385 for each extra week of shelter care including environmental enrichment [47]. 

#### 4.1.2. Admissions by Time of Year

Numbers of adult cat and kitten admissions combined were lowest in late winter and early spring in both years and highest in the warmer months (November to May). This annual cycle of admissions is consistently reported [4,8] and is due to a spring-summer influx of kittens, reflecting the cat’s breeding cycle [10,48], rather than changes in numbers of adult cat admissions [8]. Desexing programs targeted for autumn and winter, and to areas of high intake of stray and owned kittens, are recommended to prevent kittens born the previous spring and summer from producing kittens the following spring and summer. The later peak in 2011 was likely to have been a consequence of flood events in January/February of that year.

#### 4.1.3. Admissions by Desex, Feral Status, and Age

Only 27% of cat admissions in 2016, and 42% of adult cat admissions (where the cat was ≥17 weeks of age), were desexed prior to admission. This is similar to another RSPCA Queensland study of cat admissions from 2006 to 2010, in which 36% of cats and 50% of adult cats of known desex status were desexed [4]. Desexing programs are an important strategy used to reduce cat populations and stray cat admissions [9,49]. For example, free desexing programs in USA targeted to the most underserviced communities with the highest cat admissions increased desexing rates to the average for USA (90%), and reduced cat shelter admissions and euthanasia [50]. In contrast, other publications were unable to demonstrate the direct impact of desexing programs on shelter admissions due to confounding relationships between variables [50]. However, as these authors highlighted, there have been no known scientific papers or credible cases presented, either theoretical or empirical, that argue against a negative relationship between shelter intake and percentage of cats desexed in the intake area.

The analyses of cat admissions by age categories were limited in our study because 40% of the 2011 (and 12% of 2016) cat admissions did not have age recorded. Nevertheless, for those with ages recorded, the proportion of admission that were kittens (i.e., aged < 17 weeks) was greater in 2011 (57%) than in 2016 (46%) and was similar to the 54% reported from 2006 to 2008 for RSPCA Queensland [4]. The largest age category of cats admitted in 2011 was estimated to be young kittens four to seven weeks old, which represented 26% of intake, but was 17% of admissions in 2016. This reduction may have been due, in part, to diversion programs commencing in 2012, in which staff encouraged the general public intending to surrender owned or stray cats and/or their kittens to keep kittens with the dam until they reached adoption age before surrendering them. We estimate that strays represent approximately 54% to 76% of kitten admissions, and therefore desexing programs targeting stray and semi-owned cats are important to achieve further reductions in intake of kittens.

### 4.2. Outcomes

The greatest change between the two years studied was a marked increase in the percentage of admissions in which the cat was rehomed, which almost doubled (34% to 74% of intake), and a subsequent reduction in percentage euthanized, from 58% to 15%, and as a consequence, an increase in live release percentages from 39% to 81%.

#### 4.2.1. Rehomed

The number and percentage of admissions in which the cat was rehomed improved substantially for all age groups, and for kittens and cats from 4 weeks to 2 years of age, 91% were rehomed in 2016. The percentage rehomed in 2016 was lowest for cats 7 years of age and older (44%) and for young pre-weaned kittens less than four weeks (66%), although those percentages rehomed doubled from 2011 to 2016. Unweaned kittens less than 4 weeks of age or under 500gms body weight are challenging to manage in a shelter setting because of both their underdeveloped immune systems and the resources required to manage them, such as feeding every 2 to 4 h and manual stimulation of defecation and urination [51]. However, kitten nurseries devoted to caring for these at risk kittens are achieving 87% of kittens surviving and being rehomed in USA [8,14]. Between 2011 and 2016, RSPCA Queensland expanded their foster care system to manage most of these kittens, together with a small kitten nursery at the headquarters shelter. Appropriate age-specific kitten nursing protocols were implemented to increase survival probabilities of very young kittens. 

Off-site adoptions and adoption events were major contributors to the increased rehoming percentage; these accounted for only 5% of rehomings in 2011, but 22% of rehomings in 2016. This was largely due to the expanded retail partnerships with Petbarn from one to 39 stores in 2016, accounting for an increase from 86 to 1503 adoptions. Another smaller contributor to the increased rehoming percentage in 2016 was two one-day adoption events that rehomed 387 cats; neither were held in 2011. In addition to the number of cats rehomed at the adoption events, the associated marketing likely increased awareness, which might have contributed to increased numbers of in-shelter adoptions. More than half the increase in adoptions occurred in-shelter and was facilitated by a greatly expanded advertising budget from $0 in 2011 to $40,000 in 2016. Innovative advertisements were associated with higher than normal adoptions. For example, the “Geek Chic” campaign was a digital campaign that aimed to target on-line gamers (people who spend time on the Internet playing games, usually in a virtual community). It generated 2367 ‘clicks’ on the RSPCA Queensland’s cat adoption web pages by both gamers (35%) and non-gamers and RSPCA page connection audiences who clicked on the page (65%) [52]. The campaign cost $3,567 and resulted in the adoption of 611 cats. Given that every extra week in shelter care is reported to cost approximately $385/cat [47], reducing length of stay by 2 weeks would result in a saving of $470,470 for 611 cats, so this campaign was extremely cost effective. Earlier adoption also results in better welfare for the cats; less time in shelter means less stress and less risk of contracting a contagious disease.

#### 4.2.2. Reclaim

Cats reclaimed by the owner were not a contributor to the decreased euthanasia percentage, because the numbers and percentages of admissions in which the cat was reclaimed were very low in both years, and only marginally improved in 2016 (from 644 to 663 cats, or from 4.6% to 5.4% of intake). These findings are consistent with previous reports whose reclaim percentages for cats are very low both in Australia at 5% [3] and the USA at 2% [14]. Strategies aimed at improving the number of cat admissions reclaimed include mandatory microchipping in Australia [53] and USA [15]. However, a recent RSPCA Queensland study found that only 9% of incoming stray cats were microchipped, and 37% of microchips had inaccurate information (e.g., registered to a previous owner, wrong phone numbers) [54]. This highlights the need for low-cost and free microchip events targeted to locations of high stray intake, to increase the number of owned cats with microchips and correct owner contact information. In Victoria, microchipping has been mandated longer than in Queensland (2005 versus 2008), and cats also have to be registered (licensed) with the council. In Victoria, 66 of the 79 councils reported reclaim percentages on their respective websites; they averaged 13% for cats, and the top quartile of councils had 17% to 59% of cats reclaimed [55]. 

#### 4.2.3. Foster

The number of cats fostered nearly doubled from 2747 in 2011 to 4732 cats in 2016, and in 2016, 78% of cats that were fostered were rehomed within 90 days. In 2011, if a cat failed the initial behavioral assessment by displaying avoidance and/or low social behaviors, it was often deemed inappropriate for rehoming, and so was euthanized. In contrast, in 2016, many more such cats were fostered and given the opportunity to develop or demonstrate social behaviors, and were subsequently rehomed [56]. Fostering is a highly effective method for preparing cats for rehoming, and while in foster care, health problems can also be treated [17,57,58]. Foster care can provide greater environmental enrichment, with consequent beneficial effects on social behavior and health [59,60]. Utilization of temporary foster care markedly improved odds of live release for dogs, and resulted in a 70% reduction in the prevalence of major or minor health or behavior concerns compared to the prevalence in the same dogs before they were fostered [61]. Foster care is particularly valuable for improving the outcomes of older cats, which typically have longer length of stay in shelters, and as in our study, have higher euthanasia rates [62]. Foster carers, through their social networks, can also increase the number of potential adopters available. In both years of our study, about half of the cats fostered were placed in foster care by day 3 of admission to the shelter. RSPCA Queensland has been very successful in recruiting foster carers (usually volunteers), and resources directed at increasing foster care placement will subsequently further increase the number of cats rehomed. However, for cat fostering to be successful, foster carers require training and advice, resources to care for cats, and behavioral support, such as RSPCA Queensland’s foster family network [63], which is an exemplar that could be adopted by other shelters and welfare groups. Although not classed as being in foster care, some timid cats and those not coping well within the shelter environment were brought into staff offices to provide a more home-like environment and greater human interaction.

#### 4.2.4. Transferred to Rescue Groups

Cat rescue groups typically foster cats in volunteers’ homes, and their aim is to aid in the rehabilitation and rehoming of cats [57,64]. Rescue groups help to reduce the numbers of cats in shelters and expose cats available for rehoming to a larger network of potential adopters beyond the shelter [58,59]. In Queensland, over the study period, there was a growing number of cat rescue groups, but they were often at capacity, as they also receive cats from other sources (e.g., council pounds, owner surrender, etc.), hence the low number transferred to rescue groups. 

#### 4.2.5. Euthanasia

There has been a continuing reduction in euthanasia across RSPCA shelters [3], and this was reflected in significantly decreased numbers and percentages of cats euthanized from 7656 and 58% to 1826 and 15% from 2011 to 2016. In 2011, decisions to euthanize cats occurred sooner following admission (e.g., 84% of the 7656 admissions where the cat was euthanized occurred within 8 days of admission compared with 25 days for 84% of the 1826 admissions euthanized in 2016). Factors leading to the decision to euthanize included age, poor social behaviors, being classified as feral, medical reasons, and space limitations. The percentages of euthanasias for these reasons decreased from 2011 to 2016, with no cats euthanized for space limitations in 2016. The decrease in percentages and numbers euthanized was achieved by increasing capacity through an expanded foster network, by increasing distribution for rehoming through partnerships with pet shops, and by moving cats and kittens between shelters to sites of higher demand. For example, kittens continue to be born in the tropical part of the state in winter when none are being born in the southern areas and are moved south to satisfy continued demand. 

One aspect of assessing a cat’s suitability for rehoming was based on the behavior the cat displayed, assessed first by the veterinarians at admission (e.g., ‘feral’) and later whilst in the shelter environment by the behavioral assessment team (e.g., ‘timid’). Differentiating between behaviors displayed by cats that are situation-based (e.g., induced by the shelter environment or due to an underlying medical condition such as hyperthyroidism) and those that are specific to the cat is a considerable challenge, and there is a great overlap between behavioral traits. The protocols for these assessments have evolved over time, such that in 2016 more time was provided for cats to exhibit social behaviors, and a greater number of poorly socialized cats were able to be moved to foster care for socialization. As a consequence, the number of cats euthanized for behavioral reasons decreased greatly. For example, cats euthanized for behavioral (non-feral) reasons decreased from 1593 to just 261 in 2016. The most discernible decrease occurred in the number of cats that were euthanized because they were classified as feral from 1178 cats in 2011 to 132 cats in 2016 (from 8.9% to 1.1% of admissions resolved by day 90). The decision to euthanize feral cats in 2011 usually occurred within 24 h of admission, but in 2016, up to 72 h was allocated for assessment. Timid and undersocialized cats were managed by the behavior team using behavioral modification protocols, and either placed in staff offices or, where appropriate, fostered to a home where the behavior modification could be continued. The criteria for deciding whether a cat is feral are complex and controversial [65]. Over the study period, RSPCA Queensland based its decision on behavior over time and absence of identification. It is an important decision to make because under Queensland legislation, cats labelled as ‘feral’ are required to be humanely destroyed and cannot be rehomed or released (*Queensland Biosecurity Act 2014* and previously the *Queensland Land Protection (Pest and Stock Route Management) Act 2002*) [66]. Pet cats may respond with more ‘feral’ behaviors than stray cats when stressed in a shelter environment, and a minimum of 3 days is recommended before suitability for adoption is assessed [67,68,69]. However, although sufficient time must be allowed for appropriate assessment of feral status, it is extremely stressful for a truly feral cat to be kept confined in a shelter for any length of time. Strategies to reduce intake of strays will decrease numbers of feral and poorly socialized cats euthanized for behavior.

There was a marked decrease in the number of cats euthanized because they were too young. In 2011, 9.5% of the euthanized cats were kittens aged less than six weeks or under 500 g, and 5.1% were unweaned kittens. This improved greatly in 2016 to 0.4% (from 723 to 7) for kittens less than 6 weeks old or under 500g, and to 0.8% (from 393 to 15) for unweaned kittens. These decreases were the result of markedly increased availability of appropriate foster care for very young kittens, and enhanced community liaison that encouraged owners to delay surrendering kittens until they reached adoption age.

In 2016, most euthanasias were for medical reasons (68.5%), with euthanasia on humane grounds the most common reason within those (971 cats, 34.3%). The next most important medical reason for euthanasia in 2016 was because the cat had Feline Immunodeficiency Virus (FIV) (17.3% of euthanized cats, 316 cats), which was substantially higher than in 2011 (87 cats or 1.1% of euthanized cats). The seroprevalence of FIV is estimated in Australian cats between 6–15% in Perth, Western Australia [70], 6.5% to 7.5% in Sydney, New South Wales [71], and 3.6% in USA and Canada [72]. Australian studies have shown that risk factors for FIV infection include age greater than three years, entire male cats, location (e.g., inner city area have a higher seroprevalence), and ‘sick’ cats [70,71,73]. FIV prevalence is considered to be higher in Australia due to a larger population of cats with access outside the home [74]. However, it is unlikely that the increase in percentage of euthanasia because the cat had FIV was due to an increase in the prevalence of FIV in admitted cats from 2011 to 2016. Instead, cats in 2011 were more likely to be euthanized for other reasons before they were tested for FIV. In RSPCA Queensland, the Witness FeLV/FIV test was used, which has a sensitivity of 100% and specificity of 98%, and can distinguish between infected, FIV-positive, and vaccinated cats [73]. Postive tests were confirmed with polymerase chain reaction (PCR), and positive cats were usually euthanized. However, they were, in theory, available for rehoming to single cat homes with a commitment to provide specialized veterinary care when appropriate. The American Association of Feline Practitioners provides guidelines that suggest it is possible to rehome FIV-positive cats and minimize infection risk to naïve cats by implementing cat curfews and fenced outdoor shelters, keeping cats exclusively indoors, or even treating them with antiviral or immunomodulating drugs [74]. Revised adoption strategies for FIV-infected shelter cats are required if overall euthanasia percentages in shelters are to further decrease.

In 2011, 412 cats were admitted with a euthanasia request, and by implementing diversion programs, this number nearly halved in 2016 to 226 cats, and of those, owners of 111 cats provided consent to rehome where appropriate and possible. All cats admitted with a euthanasia request in 2011 were euthanized, but in 2016, 67% of the euthanasia requests with consent to rehome were rehomed, indicating that the number of cats euthanized because of euthanasia requests can be significantly reduced through diversion programs, and by providing owners with an option to consent to rehome. The adoption rate for cats whose owners gave consent to rehome was very similar to the 71% of dogs that were adopted following discussion during intake with owners surrendering dogs for euthanasia [39]. It was not determined whether owners were unaware of the options available to them, or if they misjudged the seriousness of any concerns.

In summary, the greatest future improvements in euthanasia are likely to come from decreasing intake through desexing programs, given the high proportion of intake that are strays. After implementation of a community cat desexing program in Albuquerque, USA, numbers euthanized decreased by 87% over four years, and similar improvements have been reported from other sites [42,75]. Numbers of cats in categories representing more than 10% of euthanized cats, such as poorly socialized or feral, and FIV positive cats would be expected to decrease with these programs.

#### 4.2.6. Live Release Percentage 

Overall, there was a substantial increase in the live release percentage (i.e., the sum of reclaimed, rehomed, and transferred percentages) from 40% to 81% of admissions (5360 to 9938 admissions) in which the outcome was known by 90 days after admission. Therefore, the outcomes for RSPCA Queensland shelter cats improved mainly through increased numbers of adoptions and decreased numbers euthanized including high risk groups—very young kittens and poorly socialized and feral cats.

### 4.3. Limitations

The applicability of some of our results to other populations may be affected by particular circumstances in the study period. The change from the old headquarters in Fairfield to a new, larger, purpose-built shelter at Wacol represents a limitation for some comparisons from 2011 to 2016. Significant flooding in Queensland occurred in early 2011 and impacted on shelter admissions, particularly as a number of sites were flooded (including the headquarters at Fairfield), and was reflected by a later peak for admissions in March. During the flood, RSPCA Bundaberg functioned as a flood emergency center and contributed to the higher number of admissions in 2011 (1631 cats) compared to 2016 (464 cats). 

Cat intake per 1000 residents may have been overestimated, because the intake area was assumed to only be the local government area in which the shelter was located, while cat admissions may have come from other adjacent areas. Cat age and desex status was frequently not recorded in 2011 (40% of admissions had no age, and 14% had no desex status), and cat ages were not exact but were estimates by staff, potentially biasing some of our results relating to age. Presence of microchips on admission was recorded in several fields, and the time required to retrieve and manage these data was beyond the scope of the project, so we were not able to determine whether the probability of a cat being reclaimed was associated with presence of a microchip. Capturing accurate key data such as age, microchip, and desexing status at admission for each cat is recommended; this would allow greater confidence in findings based on these data from future studies.

## 5. Conclusions

The number and percentage of admissions ending in live release for cats admitted to RSPCA Queensland shelters markedly improved in 2016 compared to 2011, by doubling the number of admissions in which the cat was rehomed. This was achieved through increased numbers of adoptions in-shelter, and through retail partnerships and one-day adoption events. Doubling the number of admissions in which the cat entered temporary foster care likely resulted in more cats being adopted, because foster care provided greater opportunities for initially poorly socialized cats to develop social behaviors, provided care for very young kittens, and increased the network of potential adopters. Further improvements will be achieved if more resources are allocated to reducing intake, such as free and low-cost desexing targeted to areas with high stray and owned cat intake into shelters, microchip and identification programs targeting areas of high stray intake, and reducing the percentage of FIV-positive cats that are euthanized. Increased funding of diversion programs to help people keep their pet is also recommended.

## Figures and Tables

**Figure 1 animals-08-00095-f001:**
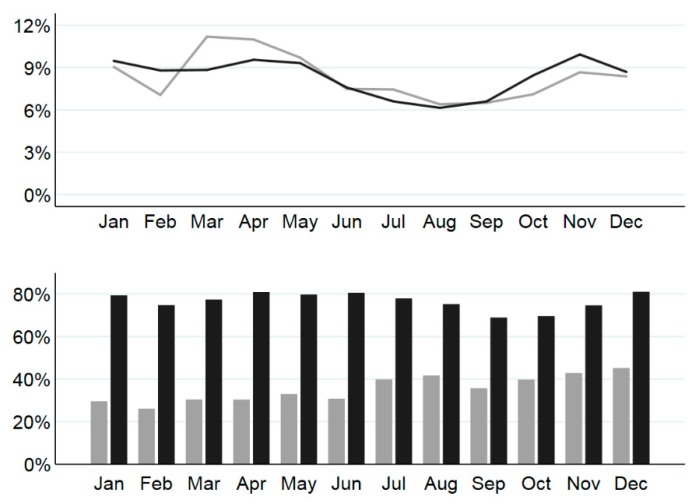
Percentages of the year’s admissions to RSPCA Queensland shelters for 13,911 cat admissions in 2011 (upper graph; grey line) and 13,220 admissions in 2016 (black line) by calendar month of admission and, in lower graph, percentages rehomed by day 90 after admission for admissions in which the cat was available to be rehomed for 12,372 admissions in 2011 (grey bars) and 10,954 admissions in 2016 (black bars).

**Figure 2 animals-08-00095-f002:**
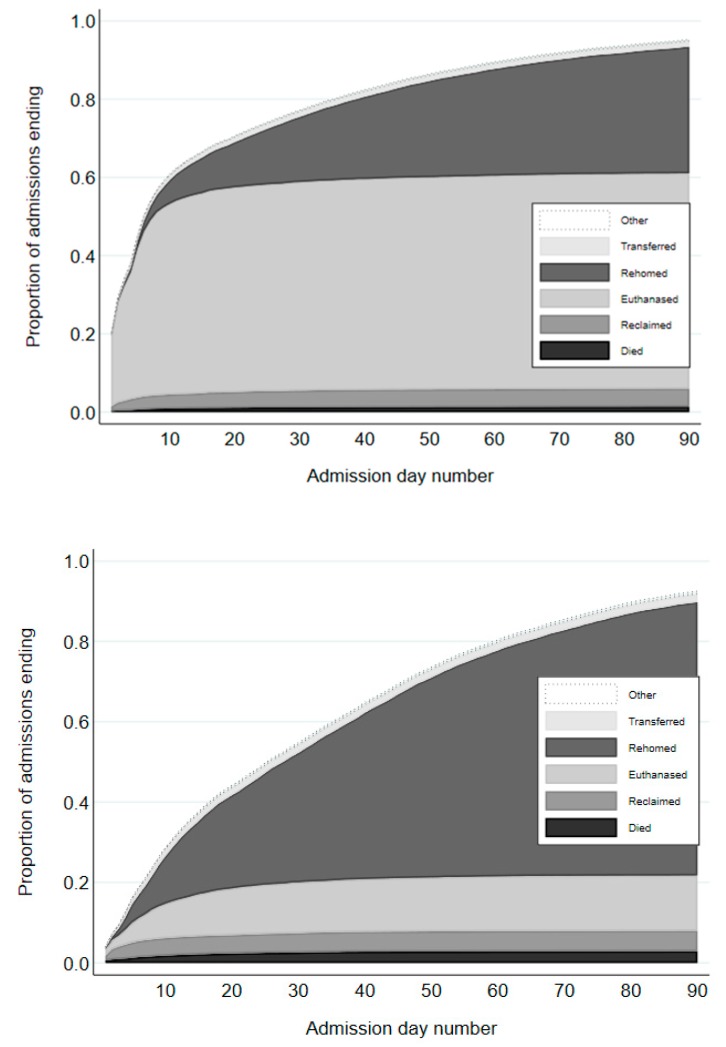
Stacked cumulative proportions of cat admissions to RSPCA Queensland shelters by outcome by day of admission (in which day 1 was the day the cat was admitted) for the years 2011 (upper) and 2016 (lower).

**Table 1 animals-08-00095-t001:** Sources of cat admissions into RSPCA Queensland shelters for the years of 2011 and 2016 and numbers and percentages rehomed by day 90 and associated risk ratios.

Source	Distribution of Admissions			% Rehomed by Day 90				
	2011 (n = 13,911)	2016 (n = 13,220)	Percentage Change ^a^	2011 (n = 12,372)	2016 (n = 10,954)	Risk Ratio ^b^	95% CI	*p*
Stray	31% (4295)	31% (4144)	−4%	31% (1246/3957)	76% (2721/3601)	0.83	0.81 to 0.85	<0.001
Owner surrender	30% (4221)	16% (2075)	−51%	44% (1774/3993)	89% (1641/1845)	Reference group		
Council contracts	20% (2715)	19% (2492)	−8%	29% (707/2404)	82% (1766/2155)	0.88	0.86 to 0.90	<0.001
Transfer in	5% (679)	12% (1553)	129%	58% (317/550)	92% (1216/1328)	1.02	1.00 to 1.05	0.024
Ambulance	5% (682)	9% (1209)	77%	16% (96/598)	43% (460/1059)	0.47	0.44 to 0.50	<0.001
Humane officer	3% (436)	4% (521)	19%	35% (131/371)	75% (349/463)	0.84	0.80 to 0.88	<0.001
Euthanasia request ^	3% (413)	2% (226)	−45%	0% (0/395)	31% (61/199)	0.19	0.15 to 0.24	<0.001
Return	1% (179)	4% (517)	189%	62% (64/104)	70% (213/304)	0.82	0.77 to 0.87	<0.001
Shelter offspring	1% (110)	4% (278)	153%					
Dead on arrival	0% (61)	1% (100)	64%					
Emergency boarding	0% (50)	1% (80)	60%					
Bequest in shelter	0% (25)	0% (20)	−20%					
Evacuation	0% (70)	0% (0)	−100%					
**Total**	**100% (13,911)**	**100% (13,220)**	−5%					

^a^ Percentage change in number of admissions (2016 relative to 2011), calculated as (4144–4295)/4295; a negative percentage change indicates a decrease in the absolute number of admissions in this category in 2016. ^b^ Estimated risk of being rehomed for the respective source, expressed as a ratio of that risk for owner surrenders, adjusted for year and disregarding interaction between source and year. ^ including euthanasia requests in which the owner gave consent for RSPCA to rehome the cat if possible and appropriate.

**Table 2 animals-08-00095-t002:** Numbers of cat admissions into RSPCA Queensland shelters by sex, desexed status, and feral status (n = 13,911 and 13,220 in 2011 and 2016, respectively) and percentages of those admissions in which the cat was rehomed by day 90 for admissions in which the cat was available to be rehomed (n = 12,372 and 10,954 in 2011 and 2016, respectively).

Factors	Distribution of Admissions		% Rehomed by Day 90	
	2011 (n = 13,911)	2016 (n = 13,220)	2011 (n = 12,372)	2016 (n = 10,954)
**Sex**				
Female	53% (6619)	50% (6414)	39% (2252/5840)	82% (4338/5271)
Male	47% (5868)	50% (6362)	40% (2082/5143)	77% (4082)/5271
Not recorded	(1424)	(444)	0% (1/1389)	2% (7/412)
**Desexed status at admission**				
Entire	77% (9165)	73% (9116)	42% (3543/8523)	82% (6727/8158)
Desexed	23% (2807)	27% (3417)	38% (789/2052)	78% (1698/2173)
Not recorded	(1939)	(687)	0% (3/1797)	0% (2/623)
**Feral status at admission**				
Not feral *	92% (12,757)	99% (13,048)	38% (4306/11,228)	78% (8374/10,801)
Feral	8% (1154)	1% (172)	3% (29 ^#^/1144) ^#^	35% (53 ^#^/153) ^#^

* Includes admissions in which feral status was not recorded as 2270 cats in 2011 and 1619 cats in 2016. ^#^ Feral status was determined at admission. Cats deemed to be feral were never rehomed; for those that were rehomed, the cat was reclassified as not feral after a period in the shelter.

**Table 3 animals-08-00095-t003:** Ages of cats on admission to RSPCA Queensland shelters in 2011 and 2016 and percentages rehomed by day 90 and associated risk ratios.

Distribution of Admissions							
Age	2011 (n = 13,911)	2016 (n = 13,220)	2011 (n = 12,372)	2016 (n = 10,954)	Risk Ratio ^#^	95% CI	*p*
<4 weeks	10% (860)	10% (1190)	42% (335/790)	66% (682/1036)	Reference group		
4 to <7 weeks	23% (1947)	15% (1720)	60% (1114/1860)	91% (1447/1595)	1.39	1.34 to 1.45	<0.001
7 to <17 weeks	23% (1945)	21% (2493)	75% (1302/1735)	95% (2156/2263)	1.49	1.43 to 1.55	<0.001
17 weeks to <1 year	18% (1531)	22% (2596)	62% (764/1227)	91% (1897/2081)	1.41	1.35 to 1.47	<0.001
1 to <2 years	7% (604)	10% (1224)	66% (301/454)	91% (851/936)	1.41	1.35 to 1.48	<0.001
2 to <7 years	13% (1106)	16% (1874)	45% (360/804)	80% (1063/1325)	1.20	1.15 to 1.26	<0.001
7 years or older	4% (311)	5% (627)	17% (38/229)	44% (189/425)	0.61	0.55 to 0.68	<0.001
Not recorded	(5607)	(1496)	2% (121/5273)	11% (142/1293)			
Total	100% (13,911)	100% (13,220)	35% (4335/12,372)	77% (8427/10,594)			

^#^ Estimated risk of being rehomed for the respective age, expressed as a ratio of that risk for cats <4 weeks old, adjusted for year and disregarding interaction between age and year.

**Table 4 animals-08-00095-t004:** Distribution of outcomes of cat admissions for all admissions to RSPCA Queensland shelters in 2011 and 2016 and the percentage of each resolved by day 90 of admission.

Outcome	2011		2016	
	Number of Admissions	% of Admissions ^1^	Number of Admissions	% of Admissions ^1^
Reclaimed	644	4.9%	663	5.4%
Rehomed	4510	34.0%	8996	73.6%
Euthanized	7656	57.8%	1826	14.9%
Transferred to rescue group	206	1.6%	279	2.3%
Died before released	193	1.5%	386	3.2%
Other (escaped or stolen)	43	0.3%	78	0.6%
In care (admission not resolved by day 90)	659		992	
**Total**	**13,911**	**100.0%**	**13,220**	**100.0%**

^1^ Percentages of those admissions that were resolved by day 90.

**Table 5 animals-08-00095-t005:** Distributions of outcomes by day 90 after admission for admissions to RSPCA Queensland shelters in 2011 and 2016 where the cat was fostered by the 30th day after admission.

Outcome	2011		2016	
	Number of Admissions	% of Admissions ^1^	Number of Admissions	% of Admissions ^1^
Reclaimed	32	1.4%	29	0.7%
Rehomed	1941	87.2%	3472	88.5%
Euthanized	192	8.6%	261	6.6%
Transferred to rescue group	14	0.5%	23	0.6%
Died before released	34	1.5%	121	3.1%
Other	12	0.5%	19	0.5%
In care (admission not resolved by day 90)	354		536	
**Total**	**2579**	**100.0%**	**4461**	**100.0%**

^1^ Percentages of those admissions that were resolved by day 90.

**Table 6 animals-08-00095-t006:** Summary of the medical and surgical reasons that cats were euthanized at RSPCA Queensland shelters for the years 2011 and 2016 for admissions in which the cat was euthanized by day 90 after admission; (**b**) summary of the age and behavioral reasons for which cats were euthanized at RSPCA Queensland shelters for the years 2011 and 2016 for admissions in which the cat was euthanized by day 90 after admission.

Reason for Euthanasia	2011		2016	
	Number Euthanized	% of All Euthanized	Number Euthanized	% of All Euthanized
**Medical/Surgical**	**2454**	**32.1%**	**1252**	**68.5%**
Cat flu	920	12.0%	128	7.0%
Humane grounds ^	434	5.6%	971	34.3%
Ringworm	328	4.3%	4	0.2%
Injured	195	2.5%	35	1.9%
Cat flu exposure	160	2.1%	1	0.1%
FIV-positive	87	1.1%	316	17.3%
Idiopathic alopecia	84	1.1%	2	0.1%
Tick paralysis	67	0.9%	2	0.1%
Dental disease	30	0.4%	2	0.1%
Malignant neoplasia	28	0.4%	32	1.8%
Neuropathy	14	0.2%	44	2.4%
Orthopedic disease	14	0.2%	18	1.0%
Obese	11	0.1%	0	0.0%
Cardiac disease	10	0.1%	18	1.0%
Ocular disorder/disease/blind	9	0.1%	4	0.2%
Failed treatment	9	0.1%	0	0.0%
Deaf	8	0.1%	0	0.0%
Paralysis	8	0.1%	5	0.3%
Demodex	7	0.1%	0	0.0%
Flea bite allergy	7	0.1%	0	0.0%
Incontinence	6	0.1%	1	0.1%
Lack of pigmentation	4	0.1%	0	0.0%
Spinal trauma	3	0.0%	9	0.5%
Chronic ear infection	3	0.0%	0	0.0%
Excessive scarring	3	0.0%	1	0.1%
Ear mites	2	0.0%	0	0.0%
Head trauma	2	0.0%	0	0.0%
Heartworm positive	1	0.0%	0	0.0%
Hyper reactivity to stimuli	0	0.0%	4	0.2%
**Age/Shelter Number Related**	**2308**	**30.1%**	**32**	**1.8%**
Shelter was full	765	10.0%	0	0.0%
Too young (i.e., <6 weeks) or under 500 g	723	9.5%	7	0.4%
Unweaned kittens	393	5.1%	15	0.8%
Geriatric	234	3.1%	10	0.5%
Too many kittens 6–8 weeks	121	1.6%	0	0.0%
Too many kittens 8–12 weeks	72	0.9%	0	0.0%
**Behavioral & Feral**	**2771**	**36.2%**	**393**	**21.6%**
Feral	1178 ^#^	15.4%	132 ^#^	7.2%
Under-socialized—not Coping	426	5.6%	89	4.9%
Timid/fearful/anxious	347	4.6%	43	2.3%
Aggression towards humans	316	4.1%	112	6.1%
Inter-cat aggression	200	2.6%	0	0.0%
Inappropriate toileting	193	2.5%	0	0.0%
Untrustworthy	39	0.5%	10	0.5%
Behavioral: general	33	0.4%	0	0.0%
Redirected aggression	13	0.2%	6	0.3%
Over grooming	10	0.1%	0	0.0%
Escape behavior	8	0.1%	0	0.0%
Compulsive, obsessive, stereotypic	3	0.0%	0	0.0%
Excessive vocalization	3	0.0%	0	0.0%
Aggression towards dogs	2	0.0%	0	0.0%
Idiopathic aggression	0	0.0%	1	0.1%
**Miscellaneous**	**123**	**1.6%**	**150**	**8.3%**
Other	97	1.3%	0	0.0%
Owner -requested euthanasia	26	0.3%	149	8.2%
Council admission	0	0.0%	1	0.1%
**Total**	**7656**	**100.0%**	**1826**	**100.0%**

^ ‘Humane grounds’ was based on severe clinical signs warranting euthanasia that were present on admission to the shelter. Therefore, the diagnosis was ‘open’. ^#^ Differences between numbers classified as feral on admission (Table 2) and numbers euthanized due to the cat being feral are due to reclassification of some cats to not being feral and others as feral after a period in shelter.

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
