# Peer review of "Changes Associated with Improved Outcomes for Cats Entering RSPCA Queensland Shelters from 2011 to 2016"

_animals, 2018, doi:10.3390/ani8060095_

Round 1

Reviewer 1 Report

The question of cats in shelters and how to improve their welfare is a highly relevant research question. Unfortunately I believe that the current manuscript, while it does give some relevant background information, does not help us to understand what to do.

Author Response

The authors thank the reviewer for their time and effort and we have made changes in response to other reviewer’s comments.

Reviewer 2 Report

This is a well written paper that presents detailed statistical analysis of the improving outcomes for cats in Queensland shelters. It acknowledges its limitations and suggests avenues for application of its findings. The conclusions are well supported by the data. The literature review is comprehensive.

Author Response

We thank the reviewer for their kind feedback. We have made changes in responses to the other reviewers which we feel has further improved the manuscript.

Reviewer 3 Report

Overall this is a well written manuscript presenting important information and evidence concerning cats that entered and exited RSPCA Queensland shelters in 2011 and 2016. One of the main comments I would like to suggest is to make the manuscript more concise and focused. To answer their main research question of interest, what changes between 2011 and 2016 contributed to the increase in proportion of rehome, two necessary associations must be assessed: 1) association with year, and 2) association with proportion of rehome. Note that having both of these two necessary associations does not imply an interaction between a factor and year on outcome but rather that this factor is associated with both year and outcome. The current presentation of the results is not in the way that the reviewer can easily use to investigate these two necessary associations. For example, Table 1 can be used to assess the first association but does not provide information for assessing the second association. To complete the investigation, readers need to use Table 5. But Table 5 is not set up for an easy assessment of these two associations either. The data for distribution of admissions by year (first two data columns) are somehow overlapped with table 1 but without % changes comparing between the two years. Association between source and rehome can be simplified by combining two years. By looking at the risk ratios across years, I would not concern too much about the interaction between source and year (discussed in lines 300-306) even it was significant (everything was significant due to large sample sizes anyway). The fact that not all RR’s were very closed to 1 indicated that source was associated with rehome. Once these two necessary associations are established, a closer evaluation of the strength and directions of these associations should follow in order to estimate potential strength and direction of the effect. Other tables have the same limitations for interpretation of the results to address the main questions of interest. Discussion also can be more focused.

By the way, I entered and analyzed the data (in table 5) using log binomial regression in STATA and the model was not converged (logistic regression, on the other hand, converged). Nonetheless, I applause the authors for their use and interpretation of risk ratios. This is important when almost all findings in their study were significant (even with a RR of 1.03). I would suggest to focus more on the factors that show a strong association.

Another main comment is that the study data were only available in two discrete years (i.e., 2011 and 2016). Thus, the phrases such as ‘from 2011 to 2016’ and ‘over the 5 years’ that imply an investigation of a continuous temporal trend during a 6-year period are misleading. Please make changes throughout the manuscript, including the title, to more accurately reflect the study time points (using phrases of, for examples, ‘comparing 2016 to 2011’ or ‘in 2011 and 2016’.

Other comments:

1.       Footnotes for feral status in table 2 should be extended to include a statement clarifying that feral status was determined at admissions. Perhaps include this under the current asterisk.

2.       Lines 275-6 in page 8: The number of RSPCA….over the five years does not read right.

3.       Lines 28-30 in page 17: Because the number of admissions in two years did not change much, I don’t think number of admissions would contribute much to changes in euthanasia/rehome in this study. Breaking down by sources of admission would be more meaningful. This also leads to my disagreement with the authors on their highlighted recommendation: Spay/neuter program. First, the study findings did not directly support the effectiveness of the program; mainly because there was no change in number of admissions comparing 2016 to 2011. Second, as mentioned in lines 44-5 in page 17, most of the admitted strays were socialized to people with low euthanasia proportion. Third, TNR is still considered illegal in most parts of the study area, which would limit the implementation of the program to reach an effective threshold for population control in stray cats.  On the other hand, it seems that the current surrender diversion strategies is an efficient way to reduce admissions and improve outcome. This is further supported by the finding that behavioral and medical reasons are the main contributors to euthanasia.

Author Response

Reviewer 3:

Overall this is a well written manuscript presenting important information and evidence concerning cats that entered and exited RSPCA Queensland shelters in 2011 and 2016. One of the main comments I would like to suggest is to make the manuscript more concise and focused. To answer their main research question of interest, what changes between 2011 and 2016 contributed to the increase in proportion of rehome, two necessary associations must be assessed: 1) association with year, and 2) association with proportion of rehome. Note that having both of these two necessary associations does not imply an interaction between a factor and year on outcome but rather that this factor is associated with both year and outcome. The current presentation of the results is not in the way that the reviewer can easily use to investigate these two necessary associations. For example, Table 1 can be used to assess the first association but does not provide information for assessing the second association. To complete the investigation, readers need to use Table 5. But Table 5 is not set up for an easy assessment of these two associations either. The data for distribution of admissions by year (first two data columns) are somehow overlapped with table 1 but without % changes comparing between the two years. Association between source and rehome can be simplified by combining two years. By looking at the risk ratios across years, I would not concern too much about the interaction between source and year (discussed in lines 300-306) even it was significant (everything was significant due to large sample sizes anyway). The fact that not all RR’s were very closed to 1 indicated that source was associated with rehome. Once these two necessary associations are established, a closer evaluation of the strength and directions of these associations should follow in order to estimate potential strength and direction of the effect. Other tables have the same limitations for interpretation of the results to address the main questions of interest. Discussion also can be more focused.

Authors’ response:

These are very good points that we had not considered when preparing the paper. We agree that disregarding the interactions is a worthwhile simplification given that the interaction terms were mostly weak and the results are easier to interpret if interaction is disregarded.

We have edited and added to Table 1, removed what was Table 4, and edited the associated Results text to focus strongly on the key points, namely whether changes in the sources and/or age categories of admissions in 2016 compared to 2011 account for the much improved outcomes in 2016. (It turns out that they did not; in fact, the increases percentage rehomed was achieved despite a decreased percentage of admissions from the two sources with highest rehoming percentages pooled (owner surrender and transfer in).

By the way, I entered and analyzed the data (in table 5) using log binomial regression in STATA and the model was not converged (logistic regression, on the other hand, converged).

Authors’ response: Strange. At the end of our responses to reviewer 3 below, we have pasted Stata code to reproduce the data and run those models. With this code, these models all converge after 6-8 iterations. This is using Stata 15 and binreg version 7.6.3 28feb2017. (The code below includes a line to check what version you have.)

Nonetheless, I applause the authors for their use and interpretation of risk ratios. This is important when almost all findings in their study were significant (even with a RR of 1.03). I would suggest to focus more on the factors that show a strong association.

Authors’ response:

We have edited the Results text to focus strongly on the key points, namely whether changes in the sources and/or age categories of admissions in 2016 compared to 2011 account for the much improved outcomes in 2016. (It turns out that they did not; in fact, the increases percentage rehomed was achieved despite a decreased percentage of admissions from the two sources with highest rehoming percentages pooled (owner surrender and transfer in).

Another main comment is that the study data were only available in two discrete years (i.e., 2011 and 2016). Thus, the phrases such as ‘from 2011 to 2016’ and ‘over the 5 years’ that imply an investigation of a continuous temporal trend during a 6-year period are misleading. Please make changes throughout the manuscript, including the title, to more accurately reflect the study time points (using phrases of, for examples, ‘comparing 2016 to 2011’ or ‘in 2011 and 2016’.

Authors’ response:

Authors agree with this recommendations and made the following changes

P1 L14, P16 L449 changed “from 2016 to 2011” to “by comparing 2011 to 2016”.

P22 L767 from “from 2011 to 2016 by “changed to “in 2016 compared to 2011,”

Other comments:

1.            Fo6otnotes for feral status in table 2 should be extended to include a statement clarifying that feral status was determined at admissions. Perhaps include this under the current asterisk.

Authors’ response:

P Table 2 footnote: Added “#Feral status was determined at admissions. Cats deemed..”

2.            Lines 265 in page 8: The number of RSPCA….over the five years does not read right.

Authors’ response:

Changed “The number of RSPCA Queensland sites from which admissions were rehomed increased by 2 over the five years, and the headquarters shelter moved from Fairfield to Wacol” to “The number of RSPCA Queensland sites rehoming cats increased by 2 when comparing 2011 to 2016. Also, the headquarters shelter site moved from Fairfield to Wacol at the end of 2011.”

3.       Lines 28-30 in page 17: Because the number of admissions in two years did not change much, I don’t think number of admissions would contribute much to changes in euthanasia/rehome in this study. Breaking down by sources of admission would be more meaningful. This also leads to my disagreement with the authors on their highlighted recommendation: Spay/neuter program. First, the study findings did not directly support the effectiveness of the program; mainly because there was no change in number of admissions comparing 2016 to 2011. Second, as mentioned in lines 44-5 in page 17, most of the admitted strays were socialized to people with low euthanasia proportion. Third, TNR is still considered illegal in most parts of the study area, which would limit the implementation of the program to reach an effective threshold for population control in stray cats.  On the other hand, it seems that the current surrender diversion strategies is an efficient way to reduce admissions and improve outcome. This is further supported by the finding that behavioral and medical reasons are the main contributors to euthanasia.

Authors’ response:

Overall, any efforts that result in less cats entering shelters will have a positive consequence of fewer cats euthanased. We agree there was no change in number of admissions comparing 2016 to 2011, and absolutely agree that looking at sources of admission are important  - at least 50%, and possibly as much as 70% of  admissions were “stray”, and most socialized to people. In 2016, only 16% were owner-surrendered cats. Therefore, future strategies that reduce stray cat intake will likely be more effective than those targeting owner-surrenders. Because RSPCA has already made substantial progress in reducing owner-surrenders, reducing “strays” admissions is the next step. In Nth America, live release rates of 95-98% for cats are being achieved in shelters where community-cat spay/neuter programs have been implemented.  We are aware that TNR programs are currently illegal in Queensland, but can be undertaken with a Biosecurity Queensland permit as a research project. Currently the two senior authors (Rand and Paterson) are involved in advanced planning such a project involving a whole zipcode. Therefore we have retained part of the text but modified it to better show the relevance.

We have changed the last sentence of the summary (P1) to: “To achieve further improvements, programs that decrease intake for both stray and owned cats would be beneficial. ”.

Reviewer 4 Report

This paper is part of an excellent series and review of factors impacting cats in Australia. It also compares the statistics from Australia with those of America, with a much larger population of cats.

Author Response

This paper is part of an excellent series and review of factors impacting cats in Australia. It also compares the statistics from Australia with those of America, with a much larger population of cats.

RESPONSE: changes made as outlined by the other reviewers.

Submission Date 29 March 2018

Date of this review 09 May 2018 09:56:24

Reviewer 5 Report

The manuscript is well written and the topic is very important to cat welfare and thus within the scope of the journal.

For purposes of clarification, there are some changes and additions to be made.

One category for euthanasia was “humane grounds”. This is not a diagnosis of a medical problem like the others from the list. Could “humane grounds” also include severe cat flu, which is another category? Please give examples for “humane grounds”, explain how it was assessed, and discuss that this is not a diagnosis of a disease…

Please generally explain how the behaviours and medical issues listed in Table 7 were assessed and by whom. Where there standardized protocols, or tests, or pre-given definitions? E.g. how was “untrustworthy” or “timid/fearful/anxious” defined and assessed? How was “hyper-reactivity” tested?

Another important question: Was each cat only assigned to one category within “behavioural & feral” and within “medical”, respectively, or could a cat appear in the statistics several times (e.g., in case of anxiety and inappropriate toiling, which is often linked, in both categories).

There is sometimes a mess with the tables. Sometimes the reference in the text does not appear in the right order.

e.g., ln 200 is referring to Table 2, but it should be Table 1?

Moreover, Table 5 is mentioned before Table 4 in the text.

Paragraph 3.1.3 should refer to the linked table

Further comments: 

Ln 25 of the discussion: There is a spelling mistake “Therefpre, was the main ….” instead of “Therefore”. And it seems the substantive of the sentence is missing.

Table 2: please indicate in the table or in the text in how many case and % feral cat status was not recorded (but included in “not feral”).

Table 3: please explain “other” also in the footnotes

Table 5 & 6: please mention the statistical tests that were performed

Table 6: category < 4 weeks old: please check the formatting -> put “reference” and “group” in the same line

Table 7: I would suggest splitting it in 2 Tables (behavioural & medical in separate tables) to make the tables fit on the page

Table A 3: the first table is shown twice, but the second time without the heading containing the rehomed/euthanized as well as the year categories. Below these two tables appears another one with the categories “reclaimed “/”transferred to rescue group”. Each table should have a separate heading, so also this one. Besides the signs of the footnotes (1, *, a) do not appear in the table itself.

Author Response

Reviewer 4:

The manuscript is well written and the topic is very important to cat welfare and thus within the scope of the journal.

For purposes of clarification, there are some changes and additions to be made.

One category for euthanasia was “humane grounds”. This is not a diagnosis of a medical problem like the others from the list. Could “humane grounds” also include severe cat flu, which is another category? Please give examples for “humane grounds”, explain how it was assessed, and discuss that this is not a diagnosis of a disease…

Authors’ response: Whilst one could argue that all euthanasias could be considered to be on humane grounds, “Humane grounds” as used in this paper means that the animal was assessed directly on admission to be suffering from an extreme condition and would be unable to be saved. To end the animals suffering they were euthanased on admission. The presenting signs with a poor prognosis included respiratory/cardiac arrest, extremely low body condition score, severe injuries, sepsis/shock etc. Hence, the actual diagnosis would have been “open” at the time of euthanasia and the decision to euthanase is based on the severity of clinical signs, i.e. poor prognosis.  Whereas, when there is defined diagnosis, such as ‘cat flu’ and the cat is euthanased e.g. due to a lack of response to treatment and therefore deteriorated enough to be euthanased, that diagnosis can be used as reason for euthanasia. Nonetheless, the reviewer makes a good point that we have overlooked. So, we have made the following change: Added the following footnote to Table 6a P14 “^ ‘Humane grounds’ is based on the severity of the presenting clinical signs on shelter admission warranting euthanasia. Therefore, the diagnosis is ‘open’.”

Please generally explain how the behaviours and medical issues listed in Table 7 were assessed and by whom. Where there standardized protocols, or tests, or pre-given definitions? E.g. how was “untrustworthy” or “timid/fearful/anxious” defined and assessed? How was “hyper-reactivity” tested?

Authors’ response:

These assessments were made by either the vets (predominantly on admission, especially in 2011) or the behavioural assessment team (usually after admission). We agree that these terms can be considered to be confusing. Some of these terms such as “fearful” are veterinary behavioursist terms, whereas others are less so (e.g. “untrustworthy”. The most important underlying assessment is 1) whether or not the cat is suitable for rehoming and 2) if the cat is displaying behaviours that question their suitability for rehoming, and are these behaviours situation based (i.e. due to being in a shelter) and not truly part of the cat’s temperament. Consequently, the following changes were made: The following sentence was added to help clarify at the start of the paragraph on page 20 L667; “One aspect of assessing a cat’s suitability for rehoming is based on the behaviour the cat displays, assessed first by the veterinarians at admission (e.g. ‘feral’) and later whilst in the shelter environment by the behavioral assessment team (e.g. ‘timid’). Differentiating between behaviors displayed by cats that are situation based (e.g. induced by the shelter environment or due to an underlying medical condition such as hyperthyroidism) and from those that are specific to the cat is a considerable challenge and there is a great overlap between behavioral traits. The protocols for these assessments have evolved over time, such that in 2016 more time was provided for cats to exhibit social behaviors, and a greater number of poorly socialised cats were able to be moved to foster care for socialization. As a consequence, the number of cats euthanized for behavioural reasons decreased greatly.”

Another important question: Was each cat only assigned to one category within “behavioural & feral” and within “medical”, respectively, or could a cat appear in the statistics several times (e.g., in case of anxiety and inappropriate toiling, which is often linked, in both categories).

Authors’ response: the reason to euthanase is recorded at the time of euthanasia. So, no, they did not appear more than once. However, we do concede that there would be overlap between the categories as suggested by the reviewer e.g. of a cat displaying inappropriate toileting would invariably be suffering from anxiety.  We added the following: P 5 M&M “The reason for euthanasia was recorded at the time of euthanasia (e.g. cat flu) and only one reason was recorded”.

There is sometimes a mess with the tables. Sometimes the reference in the text does not appear in the right order.

e.g., ln 200 is referring to Table 2, but it should be Table 1?

Authors’ response: changed to Table A1.

Moreover, Table 5 is mentioned before Table 4 in the text.

Authors’ response: reformatted the Tables/text

Paragraph 3.1.3 should refer to the linked table

Done

Further comments:

Ln 25 of the discussion: There is a spelling mistake “Therefpre, was the main ….” instead of “Therefore”. And it seems the substantive of   sentence is missing.

Authors’ response:  changed “Therefpre, was the main ….” To “the council pound and was the main’

Table 2: please indicate in the table or in the text in how many case and % feral cat status was not recorded (but included in “not feral”).

Authors’ response: added to the footnote “as 2270 cats in 2011 and 1619 cats in 2016”.

Table 3: please explain “other” also in the footnotes

Authors’ response: other was defined in the experimental section. Nonetheless, added “other (escaped or stolen)” to the table.

Table 5 & 6: please mention the statistical tests that were performed

Authors’ response: added a footnote that’s states #Generalised linear model

Table 6: category < 4 weeks old: please check the formatting -> put “reference” and “group” in the same line

Authors’ response: fixed the formatting and moved “reference” and “group” to a footnote moved.^^ Reference group

Table 7: I would suggest splitting it in 2 Tables (behavioural & medical in separate tables) to make the tables fit on the page

Authors’ response: reformatted the table.

Table A 3: the first table is shown twice, but the second time without the heading containing the rehomed/euthanized as well as the year categories. Below these two tables appears another one with the categories “reclaimed “/”transferred to rescue group”. Each table should have a separate heading, so also this one. Besides the signs of the footnotes (1, *, a) do not appear in the table itself.

Authors’ response: reformatted the table into two separate tables.

Reviewer 6 Report

Paper title:

Changes associated with improved outcomes for cats entering RSPCA Queensland shelters from 2011 to 2016

Aim:

To ‘identify changes that contributed to improved live release of cats at RSPCA QLD shelters’

General comments:

This study provides important information about the fate of cats that enter RSPCA shelters within Queensland, Australia. Overall, the quality of the study was very high. The findings were well described and the data was exhaustively scrutinised. I have no major suggestions that would improve the paper as I thought it was very impressive. I have only a few minor comments, which are primarily small editing comments:

·      The second sentence starting at L47 – isn’t that self-evident?

·      L53 – such as? Specify the types of efforts as in the next sentence you mention the consequence of these efforts.

·      L74 – Could you make it clear if the 19004 and 31178 are the total number on intakes or the number of rehomed (I think it is the first but it could be clearer)

·      L75-76 – You could remove “results from the USA……that” as the sentence reads better without it and it is still clear you mean different studies because you have the different sources referenced.

·      L79 – Mentioning that a number of studies have examined the factors that increase the probability of adoption is not useful unless you go on to mention what some of the factors noted in these studies are. This is especially important because your aim is to investigate these factors as well so you need to compare their findings to yours.

·      Following from the last point….You should also mention which region these findings come from and how that region/country may differ from Australia or Queensland.

·      L82. Can you reference the statement about the RSPCA? E.g. is this information on their website?

·      L88 – Was the aim to ‘describe’ or ‘identify’. Semantics maybe but an important difference. Perhaps it is best described as both.

·      L99 – Can you reference state legislation here about the holding periods?

·      L141 – You could refer back to Table 1 here again when mentioning the sources

·      Figure 2 looks like it needs a higher resolution, and it would be much clearer to read without the 3D effect.

·      Line numbers started again at the discussion so line numbers from here on in are the second lot of numbers.

·      L25 – ‘therefore’ has a typo

·      The discussion point about capacity at L217 I can imagine that the lower percentages of rehoming at the 90 day mark plays an important role in this aspect.

·      The term “markedly” seems to pop up a lot.

·      Table A3 – Has a typo for euthanised, and just check the table headings for the appendix tables as some need to be spread out more and have better cell alignment

Author Response

This study provides important information about the fate of cats that enter RSPCA shelters within Queensland, Australia. Overall, the quality of the study was very high. The findings were well described and the data was exhaustively scrutinised. I have no major suggestions that would improve the paper as I thought it was very impressive. I have only a few minor comments, which are primarily small editing comments:

 ·      The second sentence starting at L47 – isn’t that self-evident?

Authors’ response: we partially agree and have moved this sentence to the associated footnote.

·      L53 – such as? Specify the types of efforts as in the next sentence you mention the consequence of these efforts.

Authors’ response: P2 L53 added “such as by increasing the numbers of cats rehomed [1]”

·      L74 – Could you make it clear if the 19004 and 31178 are the total number on intakes or the number of rehomed (I think it is the first but it could be clearer)

Authors’ response: this is the number of cats rehomed. So, to clarify changed to: from 19,004 cats (29%) in 2010/2011 to 31,178 cats (56%) in 2015/2016

·      L75-76 – You could remove “results from the USA……that” as the sentence reads better without it and it is still clear you mean different studies because you have the different sources referenced.

Authors’ response: Done!

·      L79 – Mentioning that a number of studies have examined the factors that increase the probability of adoption is not useful unless you go on to mention what some of the factors noted in these studies are. This is especially important because your aim is to investigate these factors as well so you need to compare their findings to yours.

Authors’ response: added “such as ‘friendliness’ [19] and the degree of play activity displayed a by a cat [22].” To the end of the sentence mentioned above.

·      Following from the last point….You should also mention which region these findings come from and how that region/country may differ from Australia or Queensland.

Authors’ response: added “USA” to the above sentence.

·      L82. Can you reference the statement about the RSPCA? E.g. is this information on their website?

Authors’ response: added a reference.

·      L88 – Was the aim to ‘describe’ or ‘identify’. Semantics maybe but an important difference. Perhaps it is best described as both.

Authors’ response: changed ‘described’ to ‘identify’

·      L99 – Can you reference state legislation here about the holding periods?

Authors’ response: Holding period is laid down by local council and may vary between councils and there is no specific legislation. The following was added to the experimental section, “(Holding period is laid down by local council and may vary between councils).

·      L141 – You could refer back to Table 1 here again when mentioning the sources

Authors’ comment: agree but, we would still prefer to leave this information in the text.

·      Figure 2 looks like it needs a higher resolution, and it would be much clearer to read without the 3D effect.

Authors’ response:  We have replaced the figure with a higher resolution image as suggested.

·      Line numbers started again at the discussion so line numbers from here on in are the second lot of numbers.

Authors’ response: the inclusion of the landscape table has caused this. This is now corrected.

·      L25 – ‘therefore’ has a typo

Authors’ response: corrected by deleting the word.

·      The discussion point about capacity at L217 I can imagine that the lower percentages of rehoming at the 90 day mark plays an important role in this aspect.

Authors’ response:  does the reviewer mean a lower euthanasia rate at day 90?

·      The term “markedly” seems to pop up a lot.

Authors’ response: Changed in a number of locations to reduce its use as follows:

P1 L22 – changed to ‘dramatically’

P6 L237, P21, L704: changed to ‘substantially’

P11 L377, P21 L728: changed to ‘significantly’

·      Table A3 – Has a typo for euthanised, and just check the table headings for the appendix tables as some need to be spread out more and have better cell alignment

Response: corrected the spelling mistake for Table A3.